# Second-Generation Enamine-Type Schiff Bases as 2-Amino Acid-Derived Antifungals against *Fusarium oxysporum*: Microwave-Assisted Synthesis, In Vitro Activity, 3D-QSAR, and In Vivo Effect

**DOI:** 10.3390/jof9010113

**Published:** 2023-01-13

**Authors:** Paola Borrego-Muñoz, Diego Cardenas, Felipe Ospina, Ericsson Coy-Barrera, Diego Quiroga

**Affiliations:** Bioorganic Chemistry Laboratory, Facultad de Ciencias Básicas y Aplicadas, Universidad Militar Nueva Granada, Campus Nueva Granada, Cajicá 250247, Colombia

**Keywords:** Schiff base, enamine, 2-amino acids, antifungal activity, *Fusarium oxysporum*, QSAR-3D, greenhouse experiments

## Abstract

In this manuscript, the synthesis of enamine-type Schiff bases **1–48** derived from the amino acids *L*-Ala, *L*-Tyr, and *L*-Phe was carried out. Their in vitro activity and in vivo protective effect against *Fusarium oxysporum* were also evaluated through mycelial growth inhibition and disease severity reduction under greenhouse conditions. The in vitro activity of test compounds **1–48** showed half-maximal inhibitory concentrations (IC_50_) at different levels below the 40 mM range. Deep analysis of the IC_50_ variations indicated that the size of the substituent on the acetylacetone derivatives and the electronic character on the cyclohexane-3-one fragment influenced the antifungal effect. 3D-QSAR models based on atoms (atom-based approach) were built to establish the structure–activity relationship of the test Schiff bases, showing a good correlation and predictive consistency (R^2^ > 0.70 and Q^2^ > 0.60). The respective contour analysis also provided information about the structural requirements for potentiating their antifungal activity. In particular, the amino acid-related fragment and the alkyl ester residue can favor hydrophobic interactions. In contrast, the nitrogen atoms and enamine substituent are favorable regions as *H*-donating and electron-withdrawing moieties. The most active compounds (**40** and **41**) protected cape gooseberry plants against *F. oxysporum* infection (disease severity index < 2), involving adequate physiological parameters (stomatal conductance > 150 mmol/m^2^s) after 45 days of inoculation. These promising results will allow the design of novel Schiff base-inspired antifungals using 2-amino acids as precursors.

## 1. Introduction

Schiff bases and their metal complexes are widely known for their wide range of biological activities reported in the literature [1,2,3,4,5]. Although they are well-studied compounds whose synthesis has been established, meaningful attention has continually emerged over them, particularly their synergistic effect when they are present in biologically active compounds with different pharmacophores. For instance, the functionalization of the amino group of chitosan forming imine groups derived from maltol and ethyl maltol led to a series of derivatives that granted antibacterial screening, indicating that these compounds were active against *Escherichia coli* bacteria [6] (Figure 1). Novel chitosan derivatives of 2-imidazolecarboxaldehyde and 2-thiophene carboxaldehyde showed antibacterial activity, specifically the chitosan derivative of 2-thiophene carboxaldehyde, which was active against *Escherichia coli* [7]. Some chitosan-functionalized pyridine-based thiosemicarbazones and their copper(II) complexes possess substantial antiproliferative activity against Madin–Darby tumorigenic canine kidney (MDCK) MCF-7 cancer cell lines [8,9].

Some amino acids have been used to carry out the synthesis of Schiff bases [10]. *N*-salicyldienamino acid oxo-vanadium Schiff base complexes were synthesized from the reaction of sodium salicylaldehyde-5-sulfonate and some amino acids such as alanine, leucine or glycine in an aqueous medium, and leucine or tryptophan in pyridine with vanadyl acetylacetonate. The measurement of the biological activity of the complexes demonstrated an antiproliferative effect and their potential use as anticancer drugs, in addition to showing an excellent antioxidant effect [11]. The Bioorganic Chemistry Laboratory recently published the synthesis of enamines derived from *L*-Trp [12]. The Z-configured enamine was the only tautomeric form identified when aliphatic precursors were used. However, tautomeric imine forms are less stable than the corresponding enamine forms, and their isomerism depends on intramolecular hydrogen bonding and steric factors associated with the starting carbonyl precursors. In vitro biological activity tests against *Fusarium oxysporum* revealed that acetylacetone derivatives are the most active agents (IC_50_ < 0.9 mM); however, the antifungal activity could be impaired by bulky groups on ester and enamine residues [12]. These results opened up a paradigm since the synthesis of stable enamines, particularly “azaenamines” that lead to obtaining “azaenolates”, needs to be better documented and focused in the literature. In addition, they could be used for carrying out the synthesis of various heterocyclic compounds [13]. Some reports have shown the use of tandem reactions to obtain this class of intermediates that involve metal catalysts or the use of extremely strong bases such as LDA, butyllithium, or phenyl lithium, which employ severe reaction conditions [14]. Other methodologies are based on multistep reactions, such as the nucleophilic cleavage of α-amino acetals induced by TMSOTf, which can be used to prepare a wide range of substituted 1,2-amino ethers [15]. On the other hand, several new β-amino esters were prepared in two simple steps from β-keto ester derivatives and primary and secondary amines under mild conditions to carry out their subsequent hydrogenation in the presence of iridium catalysts [16].

The present study broadens the investigation of the reaction of 2-amino acids against 1,3-dicarbonyl compounds for synthesizing enamine-type Schiff bases, evaluating the use of conventional heating and microwave (MW) irradiation in a “one pot” methodology. The compounds obtained were evaluated against *Fusarium oxysporum* under in vitro mycelial growth inhibition and in vivo effect under greenhouse conditions. Furthermore, 3D-QSAR modeling was performed to provide information about the structural requirements for potentiating their antifungal activity. The results are discussed below.

## 2. Materials and Methods

### 2.1. General Information

All reagents and chemicals were commercially acquired (Merck KGaA and/or Sigma-Aldrich, Darmstadt, Germany). They were employed without additional refinement. As a result, the purity of dry solvents was sufficiently defined during purchase. The products’ progression of reactions and purifications were monitored by thin-layer chromatography (TLC) on silica gel 60 F254 plates (Merck KGaA) under detection at 254 nm. Nuclear magnetic resonance (NMR) experiments were conducted using a Bruker Avance AV-400 MHz spectrometer. TMS was used as a reference to give chemical shifts in δ (ppm). Typical splitting patterns were implemented to define the signal multiplicity (i.e., *s*, singlet; *d*, doublet; *t*, triplet; *m*, multiplet). Liquid chromatography-mass spectrometry (LC/MS) experiments were performed on an LCMS 2020 spectrometer (Shimadzu, Columbia, MD, USA), comprising a Prominence high-performance liquid chromatography (HPLC) system coupled to a single quadrupole analyzer with electrospray ionization (ESI). A Synergi column (150 × 4.6 mm, 4.0 µm) was used for analysis at 0.6 mL/min using mixtures of acetonitrile (A) and 1% formic acid (B) in gradient elution. The ESI was operated simultaneously in positive and negative ion modes (100–2000 *m*/*z* sweep), with a desolvation line temperature of 250 °C, nitrogen as a nebulizer gas at 1.5 L/min, a drying at 8 L/min, and a detector voltage at 1.4 kV. High-resolution MS (HRMS) recorded accurate mass data on a microOTOF-Q II mass spectrometer (Bruker, Billerica, MA, USA). The ESI was also operated in positive and negative ion modes (100–2000 *m*/*z* sweep), with a desolvation line temperature of 250 °C, nitrogen as a nebulizer gas at 1.5 L/min, a drying at 8 L/min, quadrupole energy at 7.0 eV, and collision energy at 14 eV. The sign of the specific optical rotations for all compounds was determined with a Jasco P-2000 polarimeter (JASCO Co., Ltd., Mary’s Court, PA, USA) in a quartz cell (1.0 cm), and the value was the mean of 10 measurements. The chemical reactions were carried out in a Discover System microwave reactor model 908,005 series DY1030 in a closed vessel controlling the temperature. FT-IR spectra were recorded on a Jasco FT/IR-6600typeA spectrophotometer (JASCO Co., Ltd., Mary’s Court, PA, USA) attached to an ATR PRO ONE accessory with an incident angle of 45 deg and resolution of 4 cm^-1^.

### 2.2. Chemical Synthesis of Compounds **1–48**

#### 2.2.1. Method A

The 2-amino acids (1 mmol) were mixed with trimethylsilyl chloride (4 mmol) in 10 mL of the desired alcohol (ROH, R = Me, Et, *i*-Pr, *n*-Bu). The mixture was heated under reflux conditions for 4 h, after which the in situ formation of the 2-alkyl 2-aminoesters was detected. Subsequently, a mixture of Et_3_N (2 mmol) and the 1,3-dicarbonyl compound (1 mmol) in the respective alcohol was added and heated under reflux conditions for 20–30 h. The reaction progress was monitored by thin-layer chromatography (TLC) until no reagents were detected. Then, the solvent was distilled off in vacuo. Finally, the residue was purified by flash column chromatography eluting with petroleum ether/ethyl acetate mixtures to give products **1–48**. Yields of the afforded compounds are listed in Table 1, and the spectroscopic data for **1–48** are presented in the Appendix A.

#### 2.2.2. Method B

The 2-amino ester compounds were prepared under microwave irradiation, mixing the reagents in the following order: 2-amino acids (1 mmol) with TMSCl (4 mmol) and the respective alcohol (1 mL). First, the mixture was heated at 120 °C for 10 min under microwave irradiation in a closed atmosphere. Then, in the same microwave tube, Et_3_N (4 mmol) and the dicarbonyl compounds (1 mmol) were added. The reaction mixture was further heated at 140 °C for 30 min under microwave irradiation. The progress of the reaction was monitored by thin-layer chromatography (TLC) until no reagents were detected. Then, the solvent was distilled off in vacuo. Finally, the residue was purified by flash column chromatography eluting with petroleum ether/ethyl acetate mixtures to obtain products **1–48**. Yields of the afforded compounds are listed in Table 1, and the spectroscopic data for **1–48** are presented in the supplementary material.

### 2.3. Antifungal Assay

Compounds were assessed against *F. oxysporum* following the previously reported 12-well plate amended-medium method to evaluate the in vitro inhibition of mycelial growth [17]. A *Fusarium oxysporum* strain (*Fox* IQB-6), recovered from a diseased *Physalis peruviana* plant, was employed for this study. IC_50_ values were calculated from log(doses) versus mycelial growth inhibition curves through non-linear regression using GraphPad Prism 9.0 for Windows.

### 2.4. 3D-QSAR Activity Structure Correlation Model Based on Atoms (Atom-Based)

The construction of the model to relate the structure and the antifungal activity was achieved by employing the atom-based 3D-QSAR approach using Phase module [18]. This approach led to determining the structural characteristics to explain and coherently understand the antifungal activity of the test compounds **1–48** according to their physicochemical properties. The molecules of **1–48** were aligned using Maestro’s standard scaffold alignment module. The SMART option was chosen, allowing the molecule’s atoms to be aligned. The compounds were randomly divided into two subsets (training and test, comprising 70% and 30%, respectively). Experimental antifungal activity assessed as inhibition of *F. oxysporum* mycelial growth (expressed as IC_50_ in M) was converted to a negative logarithm (pIC_50_ = −Log(IC_50_)) and used as an independent variable [19]. The model required four PLS (partial least-squares) components, defining a 1-Å grid space [20].

### 2.5. In Vivo Effect on F. oxysporum-Infected Cape Gooseberry Plants under Greenhouse Conditions

Cape gooseberry plants (Colombia ecotype) were employed for this in vivo assay. Thirty-day plants were inoculated by immersing the apically cut root into a previously prepared 8-day *F. oxysporum* liquid culture (1 × 10^5^ conidia/mL). Test compounds (enamines **40** and **41**) and positive control (PC, propiconazole) at IC_50_ on day 8 after inoculation were applied in the plant leaves and substrate (silty-loam soil combined with rice husk at 2:1 ratio). Hoagland’s nutrient solution was employed to fertilize plants twice weekly. The phenotypic and physiological plant responses, based on disease severity and stomatal conductance (*vide infra*), were registered at 8, 15, and 45 days after inoculation (dai) and compared to an absolute control (AC, uninfected, untreated cape gooseberry plants) and negative control (NC, *F. oxysporum-*infected, untreated plants).

### 2.6. Disease Severity Analysis

*Fusarium* wilt severity was determined 8, 15, and 45 dai. Visual examinations of the characteristic symptoms were registered, including hyponastic response, chlorosis, turgor loss in leaves, and defoliation until complete plant wilting (45 dai). These observations were quantified using the scale proposed by Moreno-Velandia et al. [21]. The disease severity index was calculated using Equation (1) described by Chiang et al. [22]:Disease severity index (DSI) = ∑[(nv)/V](1)
where n is the level of infection according to the scale, v is the number of plants present in each level, and V is the total number of plants evaluated.

The severity of the disease in each treatment was determined, and the area under the disease progress curve (AUDPC) was then calculated, following the trapezoidal integration method using Equation (2) described by Campbell and Madden (1990) [23]:(2)$$\mathrm{AUDPC}=\sum_{i=1}^{n-1}\left[\left(y_{1}+y_{i+1}\right)/2\right]\times \left(t_{i+1}-t_{i}\right)]$$
where n is the number of evaluations; *y_i_*, *y*, and *y_i_*_+1_ are the severity scale values obtained at every evaluation time; and (*t_i_*_+1_
*− t_i_*) is the time interval between evaluations.

### 2.7. Stomatal Conductance and Leaf Water Potential

Stomatal conductance was estimated on a fully expanded leaf randomly taken from the upper part of the plant using a steady-state porometer (AP4 Porometer; Delta-T Devices). Stomatal conductance (gs) was measured between 900 and 1100 h on sunny days at 8, 15, and 45 dai.

## 3. Results and discussion

### 3.1. Synthesis of Enamine-Type Schiff Bases **1–48**

To carry out the optimization process of the synthetic procedure, the reaction model whose selected precursor was *L*-tryptophan was chosen since it was the starting amino acid used in our previous study [12]. A “one-pot” strategy was initially used (method A). This strategy comprised the 2-aminoesters to be in situ obtained by the reaction between *L*-tryptophan and the respective alcohol in trimethylsilyl chloride (TMSCl) under reflux for 4 h. In order to determine the in situ formation of the intermediate alkyl esters, the reaction crude was sampled each 2 min. The sample was evaporated and redissolved in a formic acid-isopropanol mixture to be analyzed by HPLC-MS, showing the formation of each ester from the mass spectrum and, specifically, the presence of [M+H]^+^ ions. Subsequently, triethylamine (Et_3_N) was added until basic medium (Table 1). Afterward, acetylacetone was added to keep the reaction mixture under reflux until all reagents were consumed (20–30 h). Each reaction was monitored using TLC, which evidenced the formation of the same product. After the purification process, a significant increase in the yield percentages (>80%) (Table 1) was confirmed in comparison to the previous methodology [12]. Although the study model reactions had good yields, the established reaction time was quite long. Thus, a microwave (MW)-assisted protocol was then explored (method B) since this approach is widely known for its advantages in green chemistry to reduce reaction times. Hence, the reactions were carried out in closed vessels using a single-mode CEM microwave reactor. First, to obtain the respective alkyl 2-aminoester, the reagents were mixed in the following order: the amino acid *L*-tryptophan, TMSCl, and the respective alcohol (MeOH, EtOH). Next, the mixture was heated to different temperatures and reaction times, establishing that 120 °C and 10 min were the conditions in which the compound was obtained in situ with high yield. Then, Et_3_N and acetylacetone were added in the same microwave tube, and the mixture continued to be heated, varying the temperature until the reaction was completed at 140 °C for 30 min (Table 1). The TLC profiles established that the same products were obtained with better yields (>90%) than those obtained with the conventional heating methodology (Table 1). The structural elucidation of the obtained compounds demonstrated the formation of the enamines previously reported in the literature [12].

Based on the results of the reaction models using the methodologies of conventional heating (method A) and microwave irradiation (method B), we expanded the chemical space, using the amino acids *L*-Phe, *L*-Tyr, and *L*-Ala to obtain the respective alkyl esters (methyl, ethyl, isopropyl, and *n*-butyl). In addition, ethyl acetoacetate and 1,3-cyclohexanedione were also used. The yields obtained from method A are presented in light blue, and those obtained from method B are in dark yellowish green (Table 1).

The yields of compounds **1–48** varied concerning the methodology used. The low yields obtained by method A could be intended by conventional conductive heating by reflux conditions since, in this method, the vessel walls are generally heated first, and it is a prolonged process since it depends on convection currents and the thermal conductivity of the different materials, generally causing the reaction vessel temperature to be higher than that of the reaction mixture [24]. Using method A, the amino acids *L*-Tyr and *L*-Phe derivatives showed 30–70% yield percentages. In contrast, those products obtained from *L*-Ala and the dicarbonyls ethyl acetoacetate and 1,3-cyclohexanedione exhibited the lowest yields (20% to 30%), while reacting with acetylacetone obtained yields greater than 50%. The enamines derived from *L*-tryptophan showed the best yields (>70%) with the three dicarbonyl compounds (acetylacetone, ethyl acetoacetate, and 1,3-cyclohexanedione). On the other hand, the yields obtained by method B were higher than those obtained through method A, and reaction times were also reduced since microwave irradiation provides excellent control over the parameters by avoiding secondary reactions, thus improving performance and reproducibility. In addition, the dipolar coupling of microwave energy with precursors and/or solvent provides enough energy to overcome the nucleation barrier and increase supersaturation in the reaction mixture [25].

A reaction mechanism is proposed to explain the formation of the enamine tautomer (Figure 2). The first step proceeds by a nucleophilic attack of NH_2_ towards the 1,3-dicarbonyl compound to give a tetrahedral intermediate of the hemiacetalaminal type (A). Two pathways are then proposed to form imine or enamine products. Pathway 1 (green) begins with a proton abstraction from NH carried out by Et_3_N, leads to the exit of a hydroxyl group and the formation of the N=C bond, giving rise to the imine form that is later converted into an enamine by tautomeric equilibrium. In contrast, pathway 2 (purple) begins with a direct formation of the enamine through the deprotonation of the α hydrogens caused by Et_3_N, delocalizing the electrons for the formation of the double bond (NH-C=C), extending the conjugation with the group C=O and giving rise to the formation of the enamine. It is possible to affirm that pathway 1 is expected to be less favored since the acidity of the NH proton is lower compared to the α-hydrogens (RCHCONH_2_). The precursors derived from the remaining amino acids are expected to react the same way as that proposed for the *L*-tryptophan derivatives.

The low yields obtained by method A are explained by the dehydration process, which must be favored using microwave irradiation (method B) since the reaction temperature was 140 °C, which causes the evaporation of water and the equilibrium shift altogether to enamine formation. Several authors have argued that the main advantage of microwave dielectric heating over conventional heating is that the microwave-assisted approach directly couples to the dipoles or ionic molecules present in the reaction mixture, and the energy transfer occurs in less than a nanosecond, causing a rapid temperature rise. Furthermore, microwave irradiation causes volumetric heating through direct coupling of the electromagnetic field with the molecules of the reaction mixture, with little or no effect on the walls of the reaction vessel.

### 3.2. In Vitro Antifungal Assay Results

The in vitro antifungal activity against *F. oxysporum* was evaluated using the microscale amended medium assay [17] regarding the effect of test compounds **1–48** on mycelial growth inhibition. The results were expressed as half-maximal inhibitory concentration (IC_50_ in mM) for each test compound, using non-linear regression. The activity results are presented in Table 2 specifically for compounds obtained from phenylalanine, tyrosine, and alanine (i.e., **5–16**, **21–32**, and **37–48**, respectively) since the enamines afforded from tryptophan (i.e., **1–4**, **17–20**, and **33–36**) were previously reported [12]. The IC_50_ values for the Schiff bases presented high variability, ranging between 0.29 and 40.23 mM.

The Schiff bases that presented the lowest IC_50_ values (0.29–0.56 mM) corresponded to those products derived from the aromatic amino acids with acetylacetone (**2, 4**) and 1,3-cyclohexanedione (**40, 41**), while those derived from the amino acid L-Ala with ethyl acetoacetate (**29–32**) and 1,3-cyclohexanedione (**45–48**) showed the highest IC_50_ values (7–40 mM). These results suggest that the substituent size, in the case of the acetylacetone derivatives, and the electronic character in the cyclohexane-3-one fragment can influence the antifungal activity against *F. oxysporum*. On the other hand, the compounds that presented good activity were those containing methyl, ethyl, isopropyl, and *n*-butyl ester moieties, establishing that the alkyl chain probably allows interaction with the cell walls of the phytopathogen that are composed of chitin and β-1,3-glucan, responsible for the rigidity of the fungal cell wall [26]. On the other hand, the compounds that presented less antifungal activity were those derived from ethyl acetoacetate, possibly because they confer greater hydrophobicity than those obtained with acetylacetone. Przybylski et al. evaluated eight gossypol-derived Schiff bases against *F. oxysporum* radial growth. Such enamines prevailed in the enamine–enamine tautomeric forms, whose alkyl chain variations involved promising antifungal activity with inhibition percentages below 27% at 20, 1, and 0.5 µg/mL. In contrast, the imine–imine tautomer-predominated Schiff bases did not show activity against this phytopathogen [27]. These results could be compared with those obtained in this study since the synthesized Schiff bases showed promising activity by inhibiting the *F. oxysporum* mycelial growth.

### 3.3. Three-Dimensional Quantitative Structure–Activity Relationship (3D-QSAR) Based on Atoms (Atom-Based Approach)

The obtained antifungal activity based on the *Fusarium oxysporum* mycelial growth established that the synthesized enamine-type Schiff bases could be used for further antifungal development using amino acid-derived enamines. Hence, the study was extended to investigate those structural characteristics (i.e., geometric and electronic properties) of the test compounds **1–48** that influence the antifungal activity through a three-dimensional quantitative structure–activity relationship (3D-QSAR) method [28]. In this regard, the atom-based approach was then used since this method treats a molecule as a set of overlapping van der Waals spheres to encode the basic features of the local chemical structure. Hence, each atom (and thus each sphere) can fall into one of the following three categories [18]: (a) Hydrogens attached to polar atoms are classified as hydrogen-bond donors (D); (b) Carbons, halogens, and hydrogens C-H are classified as hydrophobic/non-polar (H); (c) Non-ionic nitrogens and oxygens are classified as electron-withdrawing groups (W).

Table 3 shows the statistical parameters derived from the regression for the enamine-type Schiff bases. A partial least squares (PLS) regression was performed, generating models containing one to four PLS factors. The correlation and cross-validation coefficients of the model (R^2^ = 0.85 and Q^2^ = 0.67, respectively) were statistically acceptable since the difference between these two values was not greater than 0.3. In addition, the R-Pearson value (R-Pearson = 0.82) suggested a close correspondence between the activity values (predicted and experimental IC_50_s), indicating a model with robust predictive capacity.

The F value showed the relationship between the model and the observed activity variances, indicating that this regression was statistically significant; the small variance relationship (P) indicated a higher degree of confidence in the model. The RMSE value (i.e., 0.32) represented the square error in the test set’s predictions; for a good prediction model, the RMSE value should be less than 0.5. The R^2^ Scramble (0.69) measures the degree to which molecular fields can be adjusted to random data. A value > 0.5 means that the set of variables is complete and can be adjusted. These results show that it is a statistically predictive and robust model. Thus, based on this comparative analysis of experimental and predicted activity centered on the atom-based model, pIC_50_ values for enamine-type Schiff bases were calculated. The scatterplots for the experimental and predicted activities of the evaluated Schiff bases showed a significant linear correlation for the test (R^2^ = 0.71) and training (R^2^ = 0.85) data sets (Figure 1).

3D-QSAR contour plots were analyzed to understand the effect of the spatial arrangement of structural features on the antifungal activity against *F. oxysporum*. This atom-based 3D-QSAR model uses the atom types and their occupancy in a cube grid as independent variables to fit and predict properties [29]. By default, blue cubes indicated favorable features that contribute to ligand interactions with target enzymes increasing biological activity, while red cubes indicated unfavorable features that decrease their activity [30]. The characteristics for the hydrogen-bond donors (D) showed that the favorable regions were near the NH of the pyrrole ring at indole group and a part of the enamine NH that was closer to the derivatives of the cyclohexane-3-one system, suggesting that hydrogen bond donor groups such as -OH and -NH at these positions are favorable for antifungal activity against *F. oxysporum*. In contrast, the presence of these groups near the aromatic rings of amino acids, the OH group of tyrosine moiety, and the NH group of enamines found near methylene impaired their activity (Figure 2a). The hydrophobic character is another significant component that impacts the activity. Figure 2b shows the presence of blue cubes around the aromatic part of the amino acids, esters, and enamines derived from 1,3-cyclohexanedione, which indicates that the preference of hydrophobic or non-polar groups (H) for the presence of a chain of the alkyl or phenyl groups, or a more voluminous cyclic system (cyclobutyl, cyclopentyl, or cyclohexyl) in these positions, would benefit its activity.

In contrast, the red cubes in the part of the enamines derived from ethyl acetoacetate and acetylacetone indicated that the presence of hydrophobic groups is unfavorable for activity at these positions. The presence of blue cubes around the NH of the pyrrole ring at indole and the carbonyl of the 1,3-cyclohexanedione indicated that the preference for electron-withdrawing groups such as NO_2_, CN, COOR, and SH, among others, in these positions would increase their activity. In addition, red cubes around the ethyl acetoacetate and acetylacetone derivatives and the aromatic part of the amino acids *L*-Tyr, *L*-Phe, and *L*-Trp showed a low preference for electron-withdrawing groups in these positions (Figure 2c). The contribution representation of favorable regions (Figure 2d) provides details of the relationship between structure and activity. It also offers information on the structural requirements that could be considered for the design of analogs with potential antifungal activity on *F. oxysporum*.

### 3.4. In Vivo Effect of Most Active Enamines under Greenhouse Conditions

The study was finally extended to assess the in vivo effect of the most-active enamines (**40** and **41**) on *F. oxysporum*-derived disease in cape gooseberry (*Physalis peruviana*) plants for 45 days after infection (dai) and under greenhouse conditions. Initially, a physiological parameter was measured to examine the response of *F. oxysporum*-infected plants after chemical treatment (compounds **40** and **41**, and propiconazole as positive control (PC)) compared to absolute (AC, i.e., healthy, untreated, uninfected plants) and negative (NC, i.e., *F. oxysporum*-infected plants) controls. Such a parameter includes the stomatal opening variations through stomatal conductance (SC), when plants defend themselves against pathogens, specifically fungi. In this regard, the opening reduction is linked to the protection strategy, and in the case of control, it would be linked to the plant’s proper functioning and average growth [31]. High transpiration values could generate in the leaves a risk of lowered relative water content and water potential and, in extreme cases, water embolism [32]. The leaf water deficit caused by *F. oxysporum* could rationalize the gas exchange reduction caused by stomatal closure. It has been widely demonstrated that vascular pathogens increase resistance to water movement because of the reduced diameter of conductive elements [33]. However, a transpiration rate decrease acts as a stress acclimatization mechanism in plants because low transpiration results in low water loss.

In the present study, AC and NC exhibited a typical performance, with stomatal conductance increasing and decreasing, respectively, over assay time (8 to 45 dai), as depicted in Figure 3. However, significant SC variations (*p* < 0.05) at 8 dai between controls and treatments were not evidenced. PC promoted an SC increase contrarily to the NC, which indicated an opening-recovery effect but below the AC. Compound **40** exhibited similar SC-increasing variation and, despite being below that of AC and PC at 45 dai, also indicated an opening-recovery effect. In contrast, enamine **41** promoted an SC rising at 15 dai but a reduction at 45 dai, implying a lower influence on *F. oxysporum-*derived stomatal closure at late plant development and/or disease progression [34].

*F. oxysporum*-infected plants from all treatments showed common symptoms of vascular wilt at different levels according to the incidence scale, whereas uninfected, untreated healthy plants (AC) were practically asymptomatic. In the case of the disease severity index (DSI), relevant differences were observed using the investigated treatments compared to the controls (Figure 4a). PC was the most effective treatment to reduce the *F. oxysporum*-caused symptoms in cape gooseberry plants, with ca. five-fold reduced area under the disease progress curve (AUDPC) values compared to NC. However, PC and **41** treatments had close AUDPC values, suggesting that this compound promoted a representative disease reduction, despite the significant DSI difference (*p* < 0.05) at 45 dai. In contrast, although enamine **40** promoted a better SC-recovery performance, the disease severity reduction was lower than that obtained with **41**. The observed phenotypic cape gooseberry responses substantiated the previous finding (Figure 4b) since the **41**-treated, infected plants generally exhibited a disease progression reduction and ca. 1.5-fold larger size and better plant vigor, possibly through a growth regulating or eliciting role [35,36], while **40**-treated, infected plants were similar to the AC. These compounds exhibited different in vivo performance, possibly rationalized by the structural differences based on alkyl chain at ester moiety (*n*-butyl versus methyl) and the side substituent at Cα of the amino acid residue (phenyl versus p-hydroxyphenyl). PC-treated plants exhibited lower disease severity but impacted size since this treatment afforded smaller plants. Additional studies would be required to relate the phenotypic responses observed in the treated, infected plants with the biochemical and gene expression responses and identify and define plant traits induced by treating plants with the test compounds **40** and **41**. However, these enamines could be considered promising candidates to be used as lead compounds to develop more effective controls against *F. oxysporum* in the field based on 2-amino acid-derived, enamine-type antifungals.

## 4. Conclusions

The IC_50_ values of the enamine-type Schiff bases **1–48** for *F. oxysporum* mycelial growth inhibition showed significant variability, ranging between 0.29 and 40.23 mM. The substituent size and the electronic character and the larger alkyl groups in the ester moiety influenced the activity, which could allow interaction with the cell walls of the phytopathogen. To better understand the structure–activity relationship of enamine-type Schiff bases, a QSAR-3D model based on atoms (atom-based approach) was developed, which showed a reasonable correlation and predictive power with values of R^2^ > 0.70 and Q^2^ > 0.60. Contour analysis provided information about the structural requirements that could potentiate antifungal activity for enamine-type Schiff bases to be explored in further studies to optimize the antifungal activity. In this regard, hydrogen bond donor groups near the NH of enamines, hydrophobic groups, and electron acceptors in the aromatic part of amino acids were found to be crucial for the antifungal activity. These results will allow the design of new Schiff base-inspired antifungals derived from single 2-amino acids. Indeed, the most-active compounds **40** and **41** promoted protection against *F. oxysporum*-caused disease progression under greenhouse conditions, which suggests them as lead compounds to be used in future antifungal development.

## Data Availability

Not applicable.

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
