# Peer review of "Second-Generation Enamine-Type Schiff Bases as 2-Amino Acid-Derived Antifungals against Fusarium oxysporum: Microwave-Assisted Synthesis, In Vitro Activity, 3D-QSAR, and In Vivo Effect"

_jof, 2023, doi:10.3390/jof9010113_

Round 1

Reviewer 1 Report

The article is devoted to a relevant  topic with great practical potential. The article is well-written, in accordance with all the requirements of journal.

But I have a few notes and comments:

11)     Introduction requires the presence of chemical synthesis schemes of the examples provided: lines 35-38, lines 44-48, lines 51-55, lines 61-64, lines 64-67, lines 67-69;

22)     To my surprise authors do not use IR-spectroscopy method for identification and proving structure of imines or enamines. It’s good classical method for C=N or N-C=C groups identification. I would like to see at least a few examples of IR-spectra.

33)     Scheme in the table 1 should be clarified (the right side).

44)     Authors should add pictures of 1H and 13C NMR spectra of all obtained compounds in the Supporting information.

55)      Authors should add pictures of HRMS spectra in the Supporting information.

66)     The composition and structure of all obtained compounds must be proven. There are no elemental analysis or HRMS for compounds: 5, 6, 8, 10, 13-16, 21-32, 38-40, 45-48 in the article. The data should be added.

77)     Chemical shifts of products' signals in the nmr spectra should be indicated either by increasing or decreasing. Please, correct.

Overall, I am impressed with the article and believe that it can be accepted after revision.

Author Response

Comment Reviewer 1: 

11)     Introduction requires the presence of chemical synthesis schemes of the examples provided: lines 35-38, lines 44-48, lines 51-55, lines 61-64, lines 64-67, lines 67-69;

Answer: We appreciate the referee’s comment. We included some of these examples as a new Scheme 1.

22)     To my surprise authors do not use IR-spectroscopy method for identification and proving structure of imines or enamines. It’s good classical method for C=N or N-C=C groups identification. I would like to see at least a few examples of IR-spectra.

Answer: We appreciate the referee’s comment. We included ATR-FT-IR spectra in the supplementary material for selected enamines. Unfortunately, enamines derived from L-alanine were hygroscopic, and the ATR-FT-IR spectra were not good quality, making their publication in the supplementary material difficult.

33)     Scheme in the table 1 should be clarified (the right side).

Answer: We appreciate the referee’s comment. The scheme was clarified, including the pair of Z or E geometric isomers, which could be obtained depending on the dicarbonyl compound precursor.

44)     Authors should add pictures of 1H and 13C NMR spectra of all obtained compounds in the Supplementary material.

Answer: We appreciate the referee’s comment. The graphs of 1H and 13C NMR spectra were included in the supplementary material.

55)      Authors should add pictures of HRMS spectra in the Supplementary material.

Answer: We appreciate the referee’s comment. The graphs of HRMS spectra of selected compounds were included in the supplementary material.

66)     The composition and structure of all obtained compounds must be proven. There are no elemental analysis or HRMS for compounds: 5, 6, 8, 10, 13-16, 21-32, 38-40, 45-48 in the article. The data should be added.

Answer: We appreciate the referee’s comment. Unfortunately, HRMS spectra were recorded through an external collaboration as a technical service. To low the cost of this specialized service, we selected the most representative and stable compounds to obtain HRMS spectra. Because the information obtained in NMR was coherent and correlated between all the enamines, we did not record the remaining compounds because we considered them unnecessary.

77)     Chemical shifts of products’ signals in the nmr spectra should be indicated either by increasing or decreasing. Please, correct.

Answer: We appreciate the referee’s comment. The chemical shifts were corrected in the supplementary material.

Reviewer 2 Report

In the manuscript, the authors present the preparation of 48 enamine-type Schiff base compounds by conventional heating (one-pot) and microwave irradiation methodology. The in vitro/in vivo activity study against F. oxysporum and 3D-QSAR modeling were performed.

The introduction to the manuscript is short and precise, very well focused on the problem to be addressed and the way in which it is intended to contribute. The experiments are those necessary to describe the samples and the structural implications of the designs addressed. The discussion of results and the quality of the references used as a basis for the work should be improved as described below. In relation to the validity of the references, it is noteworthy not to find citations for the year 2021, however, about 17% correspond to citations for the current year.

Based on the review conducted, I recommend that the manuscript is returned to the author for a minor revision requiring a re-review.

Page 1

Line 11: Replace “L-alanine, L-tyrosyne, and L-phenylalanine” by “L-Ala, L-Tyr, and L-Phe”  

Page 2

Line 49: Replace “its” by “their”.

Line 51: Replace “L-tryptophane” by “L-Trp”.

Page 3

Line 108: How was the presence of intermediate esters detected? Please, explain in the procedure.

Line 118: Why was the synthesis of the final compounds using MW performed with 4 mmol of TMSCl and the thermal one used only 3 mmol? Something similar reads for Et3N. Please, explain in the discussion section of the synthesis.

Page 4

Line 143: Replace “70 and 30%” by “70% and 30%”.

Line 177: Replace “0900” by “900”.

Page 5

Line 191: What is the reason for the better performance of this work, compared to the previous report (ref. 12). Is it just the time spent?

Line 199: How was the yield of intermediate esters measured, in order to establish the optimum reaction condition? Please, explain in the text.

Line 203: Replace “(>90%)” by “(> 90%)”.

Line 208: Replace “L-phenylalanine, L-tyrosyne, and L-alanine” by “L-Phe, L-Tyr, and L-Ala”.

Line 218: Replace “The amino acids L-tyrosine and L-phenylalanine” by “Using method A, the amino acids L-Tyr and L-Phe”.

Line 220: Replace “L-alanine” by “L-Ala”.

Line 221: Replace “20 to 30%” by “20% to 30%”.

Line 222: Replace “L-tryptophane” by “L-Trp”, and “>70%” by “> 70%”.

Line 241: Replace “NH” by “N+-H” and “a-hydrogens” by “a-hydrogens (CONH2)”

Page 6

In Table 1, replace “4h, 120°C, 140°C, “But” (12 times), and cC (Line 245)” by “4 h, 120 °C, 140 °C, n-Bu, and cCC”; respectively. The bond between R3 and C=O (R3-CO) in dicarbonyl compounds is not visible. Please fix.

Page 7

Line 249: Replace “140°C” by “140 °C”.

Line 253: Remove “(10-9 s)”.

Line 264: Replace “phenylalanine, tyrosyne, and alanine” by “L-Phe, L-Tyr, and L-Ala”.

Line 266: Replace “tryptophan” by L-Try”.

Line 268; Replace “1-48” by “derived from L-Phe, L-Tyr, and L-Ala”.

Page 8

Line 273: Replace “L-alanine” by “L-Ala”.

Line 274: Replace “10-40 mM” by “7-40 mM”.

Line 278: Replace “butyl” by “n-butyl”.

Line 291: CI values included in Table 2 are not discussed. Scroll to title mode “3.3. Three-dimensional quantitative structure-activity relationship (3D-QSAR) based on atoms (atom-based approach)”.

Line 310: Replace “(R-Pearson=0.82” by “(R-Pearson = 0.82).

Page 9

Line 327: Table S2 is not available in the Supporting information section.

Line 332: Replace “1-48” by “1-48”.

Line 345: Replace “tyrosine” by “tyrosine moiety”.

Line 346: Remove “(CH2)”.

Line 350: Replace “alkyl, phenyl type” by “alkyl or phenyl groups”.

Page 10

Line 357: Remove (W).

Line 359: Replace “tyrosine, phenylalanine, and tryptophan” by “L-Tyr, L-Phe, and L-Trp”.

Line 372. Please include a figure with the structure of enamine-BS 40 and 41.

Page 11

Line 398: The sentence “Although test compounds 40 and 41 did not exhibit an evident phenotypic response on healthy plants to be deduced as phytotoxicity (data not shown), some xenobiotics and fungicides can interact with various plant physiological processes, impacting the stomatal closure, as happened with other various fungicides [34].” cannot be incorporated into the manuscript since there are no supporting assays.

Line 406: Replace “days after infection (dai)” by “dai”.

Page 12

Line 426: Replace “(butyl” by “n-butyl”.

Page 13

Line 463: Replace “1-48” by “1-48”.

Referee

Author Response

Comment Reviewer 2:

Page 1

Line 11: Replace “L-alanine, L-tyrosyne, and L-phenylalanine” by “L-Ala, L-Tyr, and L-Phe” 

Answer: We appreciate the referee’s comment. We performed the recommended changes in the reviewed version of the manuscript.

Page 2

Line 49: Replace “its” by “their”.

Line 51: Replace “L-tryptophane” by “L-Trp”.

Answer: We appreciate the referee’s comment. We performed the recommended changes in the reviewed version of the manuscript.

Page 3

Line 108: How was the presence of intermediate esters detected? Please, explain in the procedure.

Answer: The formation in situ of intermediate esters was detected using HPLC-MS and identifying the [M+H] + ion for each intermediate. This explanation is now included in the section “Results and discussion.”

Line 118: Why was the synthesis of the final compounds using MW performed with 4 mmol of TMSCl and the thermal one used only 3 mmol? Something similar reads for Et3N. Please, explain in the discussion section of the synthesis.

 Answer: We appreciate the referee’s comment. We made a mistake while writing the document. The stoichiometric ratio between each amino acid and TMSCl must always be 1:4 since it is necessary to ensure an excess of TMSCl for the esterification reaction to be carried out quantitatively. This is due to the high reactivity of TMSCl with the humidity of the environment. This mistake was corrected in the manuscript.

Page 4

Line 143: Replace “70 and 30%” by “70% and 30%”.

Line 177: Replace “0900” by “900”.

 Answer: We appreciate the referee’s comment. We performed the recommended changes in the reviewed version of the manuscript.

Page 5

Line 191: What is the reason for the better performance of this work, compared to the previous report (ref. 12). Is it just the time spent?

Answer: We appreciate the referee’s comment. This manuscript was elaborated in greater depth and length, mainly for two reasons: For the manuscript published by Borrego-Muñoz, we did not have the microwave reactor, so the chemical synthesis could only be carried out using L-tryptophan as a precursor and using the reflux technique; secondly, to carry out the study under in vivo conditions in the greenhouse, it was necessary to carry out the germination of cape gooseberry seeds, their cultivation until seedlings and their transfer to the greenhouse, which involves more time and dedication in the experiments. Nevertheless, once the results were expanded, we could see that the compounds presented in this manuscript showed higher antifungal activity. Therefore, the two most active were used for the in vivo study, which has not been reported so far.

Line 199: How was the yield of intermediate esters measured, in order to establish the optimum reaction condition? Please, explain in the text.

Answer: We appreciate the referee’s comment. As we mentioned, the formation in situ of intermediate esters was detected using HPLC-MS and identifying the [M+H] + ion for each intermediate. After 10 min, we did not detect the amino acid precursor, only the signal for the formed alkyl amino ester.

Line 203: Replace “(>90%)” by “(> 90%)”.

Answer: We appreciate the referee’s comment. We performed the recommended changes in the reviewed version of the manuscript.

Line 208: Replace “L-phenylalanine, L-tyrosyne, and L-alanine” by “L-Phe, L-Tyr, and L-Ala”.

Answer: We appreciate the referee’s comment. We performed the recommended changes in the reviewed version of the manuscript.

Line 218: Replace “The amino acids L-tyrosine and L-phenylalanine” by “Using method A, the amino acids L-Tyr and L-Phe”.

Answer: We appreciate the referee’s comment. We performed the recommended changes in the reviewed version of the manuscript.

Line 220: Replace “L-alanine” by “L-Ala”.

Answer: We appreciate the referee’s comment. We performed the recommended changes in the reviewed version of the manuscript.

Line 221: Replace “20 to 30%” by “20% to 30%”.

Answer: We appreciate the referee’s comment. We performed the recommended changes in the reviewed version of the manuscript.

Line 222: Replace “L-tryptophane” by “L-Trp”, and “>70%” by “> 70%”.

Answer: We appreciate the referee’s comment. We performed the recommended changes in the reviewed version of the manuscript.

Line 241: Replace “NH” by “N+-H” and “a-hydrogens” by “a-hydrogens (CONH2)”

 Answer: We appreciate the referee’s comment. We performed the recommended changes in the reviewed version of the manuscript.

Page 6

In Table 1, replace “4h, 120°C, 140°C, “But” (12 times), and cC (Line 245)” by “4 h, 120 °C, 140 °C, n-Bu, and cCC”; respectively. The bond between R3 and C=O (R3-CO) in dicarbonyl compounds is not visible. Please fix.

 Answer: We appreciate the referee’s comment. We performed the recommended changes in the reviewed version of the manuscript.

Page 7

Line 249: Replace “140°C” by “140 °C”.

Line 253: Remove “(10-9 s)”.

Line 264: Replace “phenylalanine, tyrosyne, and alanine” by “L-Phe, L-Tyr, and L-Ala”.

Line 266: Replace “tryptophan” by L-Try”.

Line 268; Replace “1-48” by “derived from L-Phe, L-Tyr, and L-Ala”.

Answer: We appreciate the referee’s comment. We performed the recommended changes in the reviewed version of the manuscript.

Page 8

Line 273: Replace “L-alanine” by “L-Ala”.

Line 274: Replace “10-40 mM” by “7-40 mM”.

Line 278: Replace “butyl” by “n-butyl”.

Answer: We appreciate the referee’s comment. We performed the recommended changes in the reviewed version of the manuscript.

Line 291: CI values included in Table 2 are not discussed. Scroll to title mode “3.3. Three-dimensional quantitative structure-activity relationship (3D-QSAR) based on atoms (atom-based approach)”.

Line 310: Replace “(R-Pearson=0.82” by “(R-Pearson = 0.82).

 Answer: We appreciate the referee’s comment. We performed the recommended changes in the reviewed version of the manuscript.

Page 9

Line 327: Table S2 is not available in the Supplementary material section.

Answer: We appreciate the referee’s comment. We retired this table. The data are presented in Figure 1.

Line 332: Replace “1-48” by “1-48”.

Line 345: Replace “tyrosine” by “tyrosine moiety”.

Line 346: Remove “(CH2)”.

Line 350: Replace “alkyl, phenyl type” by “alkyl or phenyl groups”.

 Answer: We appreciate the referee’s comment. We performed the recommended changes in the reviewed version of the manuscript.

Page 10

Line 357: Remove (W).

Line 359: Replace “tyrosine, phenylalanine, and tryptophan” by “L-Tyr, L-Phe, and L-Trp”.

Line 372. Please include a figure with the structure of enamine-BS 40 and 41.

Answer: We appreciate the referee’s comment. We performed the recommended changes in the reviewed version of the manuscript. However, we consider that one more Figure of the chemical structures for 40 and 41, make more extends the manuscript and does not deep significatively for the discussion.

Page 11

Line 398: The sentence “Although test compounds 40 and 41 did not exhibit an evident phenotypic response on healthy plants to be deduced as phytotoxicity (data not shown), some xenobiotics and fungicides can interact with various plant physiological processes, impacting the stomatal closure, as happened with other various fungicides [34].” cannot be incorporated into the manuscript since there are no supporting assays.

Answer: We appreciate the referee’s comment. The mentioned sentence was erased of the manuscript.

Line 406: Replace “days after infection (dai)” by “dai”.

Answer: We appreciate the referee’s comment. We performed the recommended changes in the reviewed version of the manuscript.

Page 12

Line 426: Replace “(butyl” by “n-butyl”.

 Answer: We appreciate the referee’s comment. We performed the recommended changes in the reviewed version of the manuscript.

Page 13

Line 463: Replace “1-48” by “1-48”.

Answer: We appreciate the referee’s comment. We performed the recommended changes in the reviewed version of the manuscript.
